# A Systematic Review on the Role of Substance Consumption in Work-Related Road Traffic Crashes Reveals the Importance of Biopsychosocial Factors in Prevention

**DOI:** 10.3390/bs12020023

**Published:** 2022-01-25

**Authors:** Sergio Frumento, Pasquale Bufano, Andrea Zaccaro, Anello Marcello Poma, Benedetta Persechino, Angelo Gemignani, Marco Laurino, Danilo Menicucci

**Affiliations:** 1Department of Surgical, Medical, Molecular and Critical Area Pathology, University of Pisa, 56126 Pisa, Italy; s.frumento@studenti.unipi.it (S.F.); marcellopoma@gmail.com (A.M.P.); angelo.gemignani@unipi.it (A.G.); danilo.menicucci@unipi.it (D.M.); 2Institute of Clinical Physiology, National Research Council, 56124 Pisa, Italy; pasquale.bufano@ifc.cnr.it; 3Department of Neuroscience, Imaging and Clinical Sciences, “G. d’Annunzio” University of Chieti—Pescara, 66100 Chieti, Italy; andrea.zaccaro@unich.it; 4Department of Occupational and Environmental Medicine, Epidemiology and Hygiene, Italian Workers’ Compensation Authority (INAIL), 00078 Rome, Italy; b.persechino@inail.it; 5Clinical Psychology Branch, Azienda Ospedaliero-Universitaria Pisana, 56126 Pisa, Italy

**Keywords:** road traffic crashes, work-related, alcohol, drugs, medicines

## Abstract

Objective: Since many jobs imply driving, a relevant part of all road traffic crashes (RTC) is related to work. Statistics considering all crashes suggest that they are significantly associated with consumption of substances, but the root causes are not yet clear. The objective of the present paper was to systematically review the scientific literature concerning substances consumption and work-related RTC. We queried the PubMed and Scopus electronic databases according to the Preferred Reporting Items for Systematic Reviews and Meta-Analyses (PRISMA) guidelines. Articles were included if they reported all necessary data and survived a quality assessment. We selected a final sample of 30 articles from an initial pool of 7113. As hypothesized, taking any of the considered substances was found to increase the risk of work-related RTC. Descriptive statistics on work-related RTC showed a higher average positivity rate for medicines (14.8%) than for alcohol (3.02%) and drugs (0.84%). Interestingly, the impact of some medications found an unconvincing explanation in the mere occurrence of side effects, and it suggests that psychosocial and/or medical conditions could be better predictors of RTC. We therefore propose an intervention and prevention model that also considers biopsychosocial factors, for which further studies are needed in future research.

## 1. Introduction

A variety of jobs involve driving, thus implying relevant risks: at least one quarter of all road traffic crashes (RTC) are work related, causing over one-third of all occupational deaths [1]; some suggest that those numbers are underestimated in developing countries [2]. A previous systematic review on risk factors predicting the occurrence of work-related RTC mainly focused on the assessment of mechanical problems or work shifts: of note, the authors suggested that modifiable behaviors (e.g., sleepiness and fatigue) could represent the most relevant risk factors [3]. A behavior apparently easy to modify is the consumption of substances, particularly those for which the association with impaired driving ability is well known (e.g., alcohol): nevertheless, almost half of all drivers involved in RTC are found positive to at least one psychoactive substance (e.g., alcohol, drugs, or medicines; [4]), suggesting that their consumption is worth considering in work-related RTC, too. This takes on even more importance considering that the overall consumption of substances has grown by 30% from 2009 (the year following the publication of the above-mentioned systematic review) to 2019 (World Drug Report 2019, United Nations publication, Sales No.E.19.XI.8), and further increased during the pandemic [5].

Is the spread of this risky behavior (i.e., driving under influence of substances) attributable to the mere ignorance of its dangers? Or, rather, do some psychosocial factors in the work environment contribute to an underestimation of these risks? Despite representing a major issue of public health, little is known about the relationship between substance consumption and work-related RTC, and the few studies addressing this topic mainly focused on descriptive statistics rather than on the root causes of this association. A deeper understanding of this phenomenon and of the underlying risk factors could significantly improve the efficacy of awareness campaigns and prevention strategies.

The aim of this paper is to systematically review the scientific literature concerning the association between the use or abuse of alcohol/drugs/medicines and accidents at work, in particular by focusing on RTC occurred during commuting or work-related missions. Besides a mere description of this relationship, we aimed at identifying the root causes of this phenomenon by bringing in a unified biopsychosocial framework the many factors investigated separately in the literature of interest.

## 2. Methods

### Study Design and Search Strategy

This systematic review has been created according to the Preferred Reporting Items for Systematic Reviews and Meta-Analyses (PRISMA) guidelines [6]. PRISMA comprises a 27-item checklist to ensure and promote the quality of systematic reviews; this checklist is reported in Appendix A (for abstract checklist). The protocol employed in the current systematic review has been submitted for registration (ID number = 198082) to the international prospective register for systematic reviews database (PROSPERO. Available online: https://www.crd.york.ac.uk/prospero/; accessed on 24 January 2022).

The review process consisted of three phases.

Phase 1 consisted of a preliminary analysis of the literature: this was carried out in the absence of specific keywords to define the scientific databases to be analyzed, the keywords to be used, and the inclusion/exclusion criteria to be applied. The search engines used were PubMed, Google, and Google Scholar.

Based on the research question and the exploratory research of literature [3,4], the strategy was to define keywords referring to the three macro-areas of interest: (1) keywords related to alcohol, recreational drugs, and medicines; (2) keywords related to the workplace; and (3) keywords related to RTC. The keywords belonging to each category were summarized in Table 1.

Phase 2 consisted of a systematic search on the PubMed and Scopus electronic databases. An initial search was conducted in March 2020, and the final one was performed in July 2021. Defined keywords were entered in the search engines and combined using the Boolean operators “AND” and “OR”, limiting their research to titles, abstracts, and keywords. Subsequently, the research outputs were limited for language (English and Italian) and publication year (starting from 2006). The search strategy, including all keywords used and the number of studies found in each database, was analytically reported in Appendix A.

To develop an effective search strategy, we adopted the Population, Intervention, Comparison, Outcomes, and Study Design (PICOS) worksheet [6,7]. The PICOS strategy is summarized in Table 2.

Phase 3 consisted of a further in-depth selection of the papers previously included in the systematic review, aimed at extracting data of interest (i.e., data concerning work-related RTC for which substance consumption was assessed) and at confirming compliance with inclusion/exclusion criteria. During this phase, we contacted the corresponding authors of the articles reporting the information of interest only in aggregated forms (e.g., articles reporting the number of workers involved in RTC and the number of workers found positive to one or more substances, but not the intersection of these two data), asking for more detailed data.

The selected articles resulting from the PubMed and Scopus search were merged into a non-redundant database after removing duplicates. A final quality assessment was performed using a standardized tool (AXIS tool; Appendix A). Based on the data extraction, for each study, the possibility of inclusion in a meta-analysis was considered, and the corresponding authors were contacted in case raw or disaggregated data were needed; in particular, we evaluated the comparability of samples and substance assessments. The main outcomes were categorized based on their relationship with alcohol, recreational drugs, and medicines to allow a better comparison of results.

## 3. Results

### 3.1. Flow Diagram

The retrieval process from scientific literature databases has been reported in Appendix A. Items retrieved from PubMed and Scopus databases were merged in a non-redundant list containing 6109 items. Since the database query had all the keywords of the three defined macro-areas in it, the pool of selected items was large. A further selection process, illustrated in Figure 1, was applied to screen the pool: this led to its reduction to 47 studies during phase 2 and to 30 studies during phase 3.

### 3.2. Study Selection and Characteristics

Four independent reviewers (A.Z., M.L., P.B., S.F.) checked the pool of 6109 abstracts collected from PubMed and Scopus search engine outputs (excluding duplicates); any disagreement was discussed with A.G. as the arbiter. Titles and abstracts were screened, and 5856 were removed because they were not relevant for this systematic review. The remaining 253 full-text papers were checked for eligibility according to the inclusion and exclusion criteria (e.g., 40 were excluded for being systematic reviews; of these, 3 were used to better contextualize the data extracted from the retrieved studies). At the end of this process (phase 2), 47 articles that met the eligibility criteria were identified. During phase 3, aimed at extracting data of interest, a further in-depth selection was performed, and 17 other articles were removed. Specifically, four were removed because they were not related to accidents involving alcohol/drugs/medicines; six were removed because they were not related to work; five were removed because they were not related to RTC; two were removed because the corresponding authors failed to provide disaggregated data of interest, even after explicit requests. Finally, 30 articles were included (see Table 3). The quality assessment confirmed the inclusion of all selected studies (Appendix A). The heterogeneity of data extracted and their aggregation with other variables not of interest to us made it impossible to carry out a meta-analysis, despite our efforts to obtain the necessary data in disaggregated form by directly contacting the corresponding authors. In detail, too few studies investigated samples similar enough to be comparable; significant differences were found among the methods (objective or subjective) used to assess substances consumption; even when objective measures were adopted, these referred to different thresholds and legal limits.

### 3.3. Synthetized Findings

Below, for each study, we report the main findings of pertinence for the present systematic review (i.e., data concerning work-related RTC for which substance consumption was assessed), organized in relation to the substance whose impact was investigated. Out of 30 studies, 15 concerned alcohol only; 2, recreational drugs only; 2, medicines only; 7, alcohol and recreational drugs; 2, alcohol and medicines; and 2, alcohol, recreational drugs and medicines. The presence of the substance of interest could be assessed through different methods, either subjective (self-report) or objective (based on biological markers or on standardized reports contained in institutional databases); the assessment technique is reported for each article (see Table 3).

#### 3.3.1. Impact of Alcohol in Work-Related Road Traffic Crashes

Studies on the alcohol impact in work-related road traffic crashes (RTC) used both subjective and objective measures. Subjective measures consisted of Alcohol Use Disorders Identification Test (AUDIT; [8]), Alcohol Use Disorders Identification Test—Consumption (AUDIT-C; [9]), and dedicated self-report questionnaires or semi-structured interviews; objective measures consisted of blood alcohol concentration (BAC) or breath alcohol concentration (BrAC).

Since the difference between alcohol use and abuse is subjective, here, we refer to the legal limits as threshold. However, this information was not reported by all papers retrieved; among those that considered it, the legal limit was different from country to country. For these reasons, we directly specified the legal limit considered by each study when reported. Considering all studies that used an objective measure and presented disaggregated data, it was possible to calculate the main descriptive statistics concerning the positivity rates for alcohol (summarized in Table 4). Alcohol positivity rates among drivers ranged from a minimum of 1.29% [10] to a maximum of 42.67% [11]; the weighted average (WA) and its confidence interval (95% CI) of the positivity rates to alcohol: WA = 3.02%; 95% CI (2.21–4.04%).

Among 16 studies assessing alcohol assumption through objective measures, three were conducted in Australia. The first [12] merged several databases of traffic accidents using a record linkage that identified 13,124 drivers who were injured or died in a RTC over a 5-year period, on duty or during commuting; an illegal alcohol level was found in 1.36% (N = 179) of drivers. The second [13] checked the coroner’s death investigation files for crashes involving heavy vehicle drivers, finding 61 cases: only one had a BAC over legal limitation (BAC > 0.05 g/100 mL). In the third [14], data were obtained from the National Coroner’s Information System for work-related injury deaths with positive toxicological screen: out of 43 work-road deaths with positive toxicological screen, 8 had positive BAC (5 truck drivers); out of 21 commuter deaths with positive toxicological screen, 4 had positive BAC (two car drivers; one car passenger; and one pedestrian).

In Taiwan, 1286 freeway crashes involving long-distance high-deck buses were analyzed from accident reports by national police: 70% (N = 64) of the 91 drivers (corresponding to 1.2% of the total) reported as drunk (presumably based on the country legal limit; BrAC = 0.15 mg/L) were involved in fatal and injury crashes [15].

To compare injury and fatality rates in small and larger companies from British Columbia, Holizki and colleagues [16] examined a pool of 299,805 workers’ compensation claims: among vehicular incidents for which toxicology information was available (71 vehicular incidents), four drivers (5.6%) were positive for alcohol; six drivers (8.45%) were positive for a combination of drugs and alcohol.

A Turkish study [17] considered the two different limits of BAC contemplated by the country’s laws: 0 g/L for public transport, taxi, commercial, and official vehicle drivers; 0.50 g/L for drivers of private vehicles. Among 224 drivers involved in non-fatal road accidents, 191 were private vehicle drivers; 33 were public transport, taxi, commercial, and official vehicle drivers. Alcohol was detected in 61 drivers (27.2%): 60 were private vehicle drivers, the majority of whom (N = 53) had a BAC level above the legal limit (≥0.50 g/L); only one of the public transport, taxi, commercial, and official vehicle drivers was above the legal limit (0 g/L). The results showed that the higher BAC limit (≤0.50 g/L) was significantly associated with an increased risk of non-fatal accident involvement (OR = 12.29, 95% CI = 1.64–92.22), while the lower BAC limit (0 g/L) was not.

Between 2000 and 2017, the Fatality Analysis Reporting System (FARS; a compilation of all fatal crashes that occur on USA public roadways) reported 5.835 individuals fatally injured in crashes, 209 of whom were fatally injured in a work-related driving crash; among these, eight tested positive for alcohol (BAC ≥ 0.01 g/mL). The odds of being involved in a work-related fatal collision were predicted by alcohol consumption [18].

Among 33,694 Ghanaian public bus/minibus drivers involved in car crashes between 2011 and 2015, only 630 tested positive for alcohol, a factor that significantly (*p* < 0.01) increased the probability of a more severe accident [19].

In Serbia, a pool of 25,574 drivers involved in RTC during 2016 and reporting BAC levels was analyzed to evaluate the risk factors of driving under the influence of alcohol. Among commercial drivers (N = 4160), 42.67% (N = 352) of bus drivers and 32.6% (N = 1087) of truck drivers tested positive for alcohol during work-related journeys [11].

Thygerson and colleagues [20] merged three databases referring to the years 1999–2005 to identify the factors contributing to occurrence and severity of 643,647 motor vehicle crashes in Utah. Among work-related crashes, alcohol influence was reported for 1% (N = 31) of workers visiting the emergency department (N = 2330) and for 3% (N = 7) of workers hospitalized (N = 235) due to more severe injuries.

Three French national databases reported 179,269 drivers involved in an injurious crash between July 2005 and December 2015 [21]. In the whole sample of injurious crashes, 24% (N = 43,012) occurred during commuting; of these, 1.48% (N = 638) had a BAC level above 0.05. Similarly, in the whole sample of injurious crashes, 12% (N = 21,490) occurred during work missions; of these, 0.93% (N = 200) had a BAC level above 0.05.

Chen and colleagues [22] analyzed the US Statewide Integrated Traffic Records System (SWITRS) to find truck-involved collision data occurred in Los Angeles from January 2010 to December 2018: among 21,258 truck drivers involved in a RTC, 7.26% (N = 1544) tested positive for alcohol.

Overall, 1800 traffic fatalities that occurred between 2004 and 2012 were identified from the USA Fatality Analysis Reporting System (FARS): 2.1% (N = 38) of all fatalities were characterized as both work-related and alcohol-impaired [23].

Based on the Texas Crash Records Information System (CRIS), Li and colleagues [10] analyzed 85,184 large truck RTC and found that 1.29% of the drivers (N = 1103) were under the influence of alcohol: 241 were classified as involved in serious RTC (incapacitating or fatal crashes), and 862 were classified as involved in non-serious RTC (all crash types excluding incapacitating or fatal crashes).

Yuan and colleagues [24] used the Fatality Analysis Reporting System (FARS) to analyze risk factors associated with truck-involved fatal crash severity: among the whole sample of 15,506 truck drivers, 2.4% (N = 372) were drunk driving.

Among 38,240 car drivers and motorcyclists involved in a RTC between 2001–2011 in Australia, 10.2% (N = 3888) were travelling for work-related reasons [25]. Among 2530 car crashes that happened during work-related travel, 2.2% of car drivers (N = 56) were drunk driving; among 1358 motorcyclists crashes that happened during work-related travel, 1.0% of motorcyclists (N = 13) were drunk driving.

Seven studies used subjective measures to assess alcohol assumption. Among 1133 Brazilian bicycle commuters, a semi-structured survey found that riding immediately after alcohol ingestion represents a risk factor only if included in a set of extremely imprudent behaviors (*p* = 0.005) [26]. Contrary to this result, a semi-structured questionnaire administered to 712 Ethiopian taxi drivers revealed that a history of alcohol use was an independent predictor (AOR = 1.51; CI = 1.002–2.28; *p* = 0.049) of RTC involvement [27].

In a sample of 227 Israelian commercial (i.e., bus or truck) drivers, the severity of alcohol misuse—as assessed by AUDIT—was significantly related to the self-reported number of accidents that occurred in the past year [28]; the strength of such correlation was higher for accidents in which the driver only was injured (r = 0.28; *p* < 0.01) than for accidents in which multiple people were injured (r = 0.16; *p* < 0.05) or in which fatalities occurred (r = 0.19; *p* < 0.05).

To obtain prevalence data on the use of alcohol and other drugs among Belgian workers, a questionnaire including AUDIT-C was administered to 5367 workers: among 4197 of those reporting to have used alcohol in the last year, 25 (0.6%) were involved in RTC on journeys to or from work [29].

To investigate the factors associated with truck crashes, Thiese and colleagues [30] administered questionnaires for the self-report of crashes, demographics, psychosocial factors, and other elements (such as alcohol consumption) to 797 truck drivers: among the 469 (58.9%) of those reporting alcohol consumption, 195 had been involved in a crash in their lifetime. The results showed that alcohol use significantly increased the risk of having crashes (AOR = 1.35; CI = (1.01, 1.81)).

In a sample of 253 Italian truck drivers, Papalia and colleagues [31] administered a survey to investigate some variable (e.g., sleep disturbances, visuo-motor performances, assumption of medications) that might increase the risk of work-related injuries and of RTC: use of alcohol was not significantly correlated with the risk of RTC (*p* = 0.9).

Two studies were conducted in Ghana. In the first one, Konlan and colleagues [32] administered a questionnaire to commercial motorcyclists to determine the prevalence and pattern of motorcycle crashes. Among 114 commercial motorcyclists, 64% (N = 73) were involved in at least one crash, 39 of whom (34.2% of total sample size) had a history of alcohol use. The results showed that the prevalence of crashes was higher among the commercial drivers who drank alcohol than among those who did not (χ2 = 33.294, *p* < 0.05). In the second study, Poku and colleagues [33] assessed driver, vehicular, and road-related factors associated with RTC by administering a semi-structured questionnaire to 227 drivers. Overall, 55.5% (N = 126) were involved in at least one crash, 31 (24.6%) of whom had drunk before driving. The results showed that alcohol use increased the risk of having RTC (cOR = 2.42; CI = (1.17, 5.00)).

#### 3.3.2. Impact of Recreational Drugs in Work-Related Road Traffic Crashes

Studies on the drug’s impact in work-related road traffic crashes used both subjective measures consisting of self-report questionnaires, semi-structured interviews, surveys, and objective measures consisting of toxicological screens.

Seven studies used objective measures to assess drug consumption.

Considering all studies that used an objective measure and presented disaggregated data, it was possible to calculate the main descriptive statistics concerning the positivity rates for recreational drugs (summarized in Table 4). The positivity rates for recreational drugs among drivers involved in a RTC ranged from a low of 0.6% [10] to a high of 21.33% [14]; the weighted average (WA) and its confidence interval (95% CI) of the positivity rates to alcohol: WA= 0.84%; 95% CI (0.30–1.84%).

Brodie and colleagues [13] examined an Australian sample of 91 heavy vehicle drivers killed in RTC: among the 61 dead drivers considered, 16.3% (N = 10) were found positive to substances (only stimulants = 6; only cannabis = 2; cannabis and stimulants = 2).

Consulting the Fatality Analysis Reporting System (FARS) database, Gates and colleagues [34] analyzed 10,190 fatal crashes occurred between 1993 and 2008 and involving USA truck drivers tested for stimulant use: 3.7% of them (N = 372) were positive for stimulants (12 different types), the use of which was also found to increase the risk of performing an unsafe driving action in a fatal crash (AOR = 1.78; 95% CI = 1.41–2.26).

To compare injury and fatality rates in small and larger companies from British Columbia, Holizki and colleagues [16] analyzed the vehicular incidents for which toxicology information was available (71 vehicular incidents): 18.3% (N = 13) of drivers were positive for one drug only; 8.45% (N = 6) of drivers were positive for a combination of drugs and alcohol.

To estimate the prevalence of work-related deaths associated with drugs, McNeilly and colleagues [14] used data obtained from Australian National Coroner’s Information System (NCIS) for work-related injury deaths that had positive toxicological screens: among 43 work-road deaths with positive toxicological screen, 5 (11.6%) had cannabis detected at levels indicating recent use, and 5 (11.6%) had amphetamine type-stimulants (ATS) detected; out of 21 commuter deaths with positive toxicological screen, 5 (23.8%) had cannabis detected at levels indicating recent use (three car drivers; two motorcyclists), and 2 (9.5%) had ATS detected.

Rudisill and colleagues [18] consulted the Fatality Analysis Reporting System (FARS) to analyze 5835 cases of individuals fatally injured in crashes occurred between 2000 and 2017, 209 of whom were fatally injured in a work-related RTC: 28 (13.39%) drivers were found positive for drugs.

Based on the Texas Crash Records Information System (CRIS), Li and colleagues [10] analyzed 85,184 large truck crashes and found that 0.36% of the drivers (N = 305) were under the influence of drugs: 99 were classified as serious (incapacitating or fatal) crashes, and 206 were classified as non-serious (all crash types excluding incapacitating or fatal) crashes.

Yuan and colleagues [24], using the Fatality Analysis Reporting System (FARS), analyzed risk factors associated with truck-involved fatal crash severity: among the whole sample of 15,506 truck-drivers, only 1.2% (N = 186) were under drug influence while driving.

The remaining two studies used subjective measures to assess drug use.

Lambrechts and colleagues [29] administered a survey among 5367 Belgian workers to assess the job-related consequences of drug use: among the 403 of them reporting drug use in the past year, 4 (1.1%) were involved in RTC to or from work.

Wadsworth and colleagues [35] investigated the association between cannabis use, injuries, and accidents in England by administering a questionnaire to 30,000 people: RTC related to work were 2801, for which it was estimated that cannabis use tripled the risk of injury (OR = 3.01; CI = 0.89–10.17).

#### 3.3.3. Impact of Medicines in Work-Related Road Traffic Crashes

As for the other substances and the impact of medicines, studies used both subjective measures (self-report questionnaires, semi-structured interviews, or surveys) and objective ones (toxicological screens). Even if presumably based on medical diagnoses, it cannot be excluded that the drug’s assumption was a patient’s initiative when not otherwise specified (as in [21]). Three studies adopted objective measures (i.e., toxicological screens, medical prescriptions) to assess the use of medicines.

Considering all studies that used an objective measure and presented disaggregated data, it was possible to calculate the main descriptive statistics concerning the positivity rates for medicines (summarized in Table 4). The positivity for medicines among drivers ranged from a minimum of 1.03% [36] to a maximum of 43.75% [14]; the weighted average (WA) and its confidence interval (95% CI) of the positivity rates to alcohol: WA= 14.8%; 95% CI (10.74–19.8%).

Reguly and colleagues [36] examined the impact of opioid analgesics (OA) in a large (N = 65,361), cross-national (i.e., all 50 states of USA, the District of Columbia, and Puerto Rico) sample of truck drivers involved in fatal crashes during a 15-year period (1993 to 2008): among truck drivers tested for drugs (N = 10,190, corresponding to 15.6% of the total), male truck drivers using OA had greater odds of committing unsafe driver actions (OR = 2.80; CI = 1.64–4.81).

McNeilly and colleagues [14] checked data obtained from the National Coroner’s Information System to estimate the prevalence of medicines-related occupational deaths in Australian workers: 43% (N = 9/21) of commuters and 44% (N = 19/43) of workers involved in a work-road death had positive toxicological screen for one or multiple medicines (i.e., opioids, benzodiazepines, antidepressants, and anticonvulsants).

Bourdeau and colleagues [21] investigated the association between exposure to ten classes of medicines and the risk of being responsible for a RTC, extracting data from three French national databases. Overall, 179,269 drivers involved in an injurious crash between July 2005 and December 2015 were identified: of these, 24% (N = 43,012) were commuting at the time of crashes, 17.9% of whom (N = 7689) had been exposed to at least one medicine; 12% (N = 21,490) were on a work-related mission at the time of crashes, 15% of whom (N = 3213) had been exposed to at least one medicine. Among the ten classes of medicines investigated, the higher risk was associated with the assumption of: antiepileptics (AOR = 1.63; CI = (1.24, 2.15)), psycholeptics (AOR = 1.19; CI = (1.03, 1.39)), or psychoanaleptics (AOR = 1.37; CI = (1.17, 1.60)) for the commuters; and antiepileptics (AOR = 1.59; CI = (1.01, 2.51)), other nervous system drugs (AOR = 2.04; CI = (1.35, 3.07)), or psychoanaleptics (AOR = 1.35; CI = (1.02, 1.78)) for the drivers on a work-related mission.

Two studies used subjective measures to assess medicines consumption.

To determine the relationship between medicines consumption and the occurrence of RTC among Iranian commercial (truck and bus) drivers, 323 of them were selected: the consumption of the medicines Gemfibrozil (used to reduce cholesterol) and Glibenclamide (used to treat type 2 diabetes) significantly increased the incidence of RTC (respectively, OR = 1.8, *p* = 0.010; OR = 2.2, *p* = 0.002) [37].

Papalia and colleagues [31] administered a survey to 253 truck drivers of public Italian transport, investigating variables (e.g., sleep disturbances, visuo-motor performances, assumption of medications) possibly increasing the risk of work-related injuries and of road accidents: use of antihistamines and benzodiazepines was found to significantly correlate with the risk of RTC (*p* = 0.003).

#### 3.3.4. Synthetized Findings from Studies Reporting Aggregated Data of Interest

Of all studies selected, two failed to specify data prevalence for each substance considered, even after explicit request sent to the corresponding author; data of interest were reported in an aggregated form only.

In the first study [38], to identify the factors that influence the frequency and severity of rear-end crashes in work zones, 2481 rear-end crashes were analyzed: 1.45% of drivers (N = 36) were found positive to either alcohol, drugs, and/or medicines.

In the second study [39], to investigate the injury severity of rear-end crashes involving large trucks, 7976 rear-end crashes were analyzed: 0.45% of drivers (N = 36) were found positive for alcohol and/or drugs.

#### 3.3.5. Impact of Psychological, Medical, and Social Factors in Work-Related Road Traffic Crashes

Some of the studies selected for this systematic review also provided information regarding the impact of psychological, medical, and social factors in work-related RTC linked to substance assumption.

Vulnerability to alcohol consumption in the workplace was significantly increased by job-related stressors (role conflict and supervisory abuse) combined with weak social monitoring of drinking behaviors [28]; this phenomenon was significantly augmented in companies (1) with ≥ 200 workers (21.8% vs. 17.7%; *p* = 0.001), (2) in the government and education sectors (*p* < 0.001), employing workers with (3) a higher level of education (29.3% vs. 11.2%; *p* < 0.001), or (4) ≤ 35 years (21% vs. 17.8%; *p* = 0.004) [29]. Interestingly, a self-reported history of alcohol use predicted work-related RTC [27] more strongly than the ingestion of alcohol preceding a ride [26].

Regarding drug assumption, a higher risk was significantly associated with a low level of education and with living alone [29].

Some of the studies investigating the impact of medicines on RTC also looked into the reasons underlying their consumption in truck drivers: in particular, the spread of stimulants was reported to counterbalance the fatigue due to high-stress working conditions [34], whereas the medicines whose consumption correlated with an increased crash probability were prescribed to treat medical conditions (e.g., diabetes) linked to a sedentary lifestyle [37]. Moreover, to explain the effects of some classes of medicines (e.g., anti-epileptic) on RTC, Bourdeau and colleagues [21] hypothesized a stronger relation with the condition itself (epileptic seizures) than with their intake.

## 4. Discussion

The main purpose of the present review was to systematize the scientific literature linking the assumption of alcohol, drugs, and medicines with work-related road traffic crashes (RTC), since the growing attention that this topic is receiving by many private and public institutions (e.g., social insurance systems).

Based on strict inclusion criteria (Table 2) and quality assessment (Appendix A), we selected 30 articles addressing the topic of our interest. The heterogeneity of extracted data and the unavailability of raw data (even after explicit requests to the authors) made it impossible to perform a meta-analysis. Considering the number-per-year and geographic source of the included studies, it clearly emerges that the publication rate for this research line is rapidly increasing, being the object of a worldwide investigation.

This growing attention is not equally shared among the various substances: most of the papers (26) focused on the role of alcohol in work-related RTC, followed by recreational drugs (11) and medicines (6) (some studies considering more substances; see Table 3). Studies comparing positivity rates to more substances reveal that this greater interest for alcohol is coherent with its higher consumption: however, drugs were associated with higher fatality rates with respect to alcohol (see Table 3). Further noteworthy information concerns the different consumption of substances during work-related and non-work-related journeys: both alcohol and drugs are less used during work-related journeys, but this reduction is halved for drugs [18].

Among studies investigating the impact of alcohol, we found a wide agreement on the conclusion that consumption of alcohol significantly increases the risk of work-related RTC, even if the distinction between use and abuse was based on different threshold. Considering that the descriptive statistics concerning the positivity rates to alcohol (summarized in Table 4) are based on samples differing in relevant characteristics, the odds of testing positive to alcohol after an accident, as averaged over the studies providing the needed information, were 3.02%. While the larger number of studies concerning alcohol makes this value more reliable than that referring to drugs and medicines, the wide range (1.29–42.67%) of positivity rates across papers suggests that this measure was heavily affected by differences in the samples considered. Interestingly, most of the studies detailing the severity of injuries found a positive correlation with the severity of alcohol consumption (see Table 3). However, it is still debated if alcohol represents an independent predictor of work-related RTC; while Asefa and colleagues [27] found history of alcohol consumption to be an independent predictor of work-related RTC, Bacchieri and colleagues [26] found that riding after alcohol ingestion represented a risk factor only if considered in the wider context of imprudent driving behaviors.

In the scientific literature concerning the impact of substances on work-related RTC, recreational drugs represent the most investigated substance class, after alcohol; such substances are typically classified as either stimulants (e.g., amphetamines) or depressants (e.g., cannabis) based on their biological mechanisms, even if the subjective experience can be more nuanced. As for alcohol consumption, we found a significant agreement on the positive correlation between recreational drugs assumption and an increased risk of work-related RTC. However, none of the included studies claimed recreational drugs to be an independent risk factor. With the premise that the descriptive statistics concerning the positivity rates to any type of drug (summarized in Table 4) are based on samples differing in relevant characteristics, the odds of testing positive to any type of drug after an accident, as averaged over the studies providing the needed information, were 0.84%. This value is lower than that referring to alcohol or medicines; however, the small number of studies (7) considered and the wide range (0.6–21.33%) of positivity rates across papers suggest that this information should be taken with caution.

Medicines represent the less-studied substances in the scientific literature considered for the present review, having been investigated in only 6 out of 30 papers. However, only one study specified that the medicines’ consumption followed a prescription; we encourage future studies to specify this information. Considering that the descriptive statistics concerning the positivity rates to any type of medicines (summarized in Table 4) are based on samples differing in relevant characteristics and on three studies only, the odds of testing positive after an accident, as averaged over the studies providing the needed information, were 14.8%. Despite the wide range (1.03–43.75%) of positivity rates obtained from only three studies, the magnitude of the odds (respectively, five and fifteen times bigger than the odds of alcohol and drugs) suggests that medicines are the most dangerous substance when driving. Considering this evidence, the small number of studies investigating the effect of medicines on driving seems undeserved; future studies should fill this gap and clarify whether this association is directly linked to the medicines or to the underlying condition. While is not surprising that opioid analgesics increase the risk of unsafe driver actions ([21,36]; however, Monárrez-Espino and colleagues [40] found unconvincing evidence of a direct association between opioids analgesics and RTC), the positive correlation between work-related RTC and the consumption of medicines that do not have neat psychoactive effects (e.g., anticonvulsants, as reported by [14]; antihistamines, as reported by [31]; medicines that reduce cholesterol, as reported by [37]) deserves a deeper analysis. The possibility that such correlation only relies on the side-effects of these substances is not conclusive, since these effects occur in a minority of cases and are occasionally described after the consumption of a variety of medicines (e.g., no relation with RTC was found for one of the medicines mostly used by patients with diabetes, insulin, the side-effects of which are consistent with those of medicines correlated with an augmented risk of RTC; [37]). Rather, it should be taken into consideration the possibility that the main risk factor was the psychological and/or medical condition for which the medicine was prescribed, as suggested also by Bourdeau and colleagues [21].

This hypothesis was not directly nor specifically addressed in any of the included papers: however, considered as a whole, they provide data (summarized in Section 3.3.5) that support their contextualization in a more complex framework underlying the impact of substance consumption in work-related RTC. Coherently with the model developed by Engel [41], the few studies investigating also psychological, medical, and social factors linked to work-related RTC found that consumption of substances could be contextualized in a situation of biopsychosocial discomfort; here, the root causes of risky behaviors should be searched for (of which driving under influence of substances represents a notable example). The assessment of biopsychosocial factors could be equally or more effective than the direct assessment of substance consumption; for example, assessing a history of substance abuse predicted work-related RTC more strongly [27] than directly assessing the substance consumption [26].

In this perspective, solving an overt problem (the consumption of substances) could just lead to the adoption of other risky behaviors as a compensation (Figure 2). Let us consider the example of a transport company willing to eradicate substance consumption among its employees by testing it before each ride: even not considering the economical and practical efforts, the intervention could be perceived as unreasonably punitive and demeaning for the workers. This could possibly increase their biopsychosocial discomfort, paradoxically increasing their desire to indulge in other risky behaviors. On the other hand, a more feasible (and better accepted) intervention could focus on a continuous biopsychosocial assessment of the working context (Figure 2), providing free medical and psychological help for employees.

Concluding, with respect to the only previous review addressing a similar topic [3], the present work (1) updated the epidemiological data on the association between substance consumption and work-related RTCs, (2) summarized descriptive statistics on positivity rates, and (3) focused on the underlying biopsychosocial factors, thus proposing a model of intervention. In this framework, a systematic review and meta-analysis concerning all work accidents related to substance consumption could provide useful information to directly target the biopsychosocial factors that may lead to RTC. In absence of a deeper investigation of the context underlying the consumption of substances, its detection could represent just the tip of the iceberg.

Among all road traffic crashes (grey circle), work-related ones (red slices) represent about a quarter of the total [1]; consumption of substances (white dots pattern) was reported in about 50% of all traffic crashes [3]. Current interventions (represented by light-green arrows) improve risk awareness (blue square 1) and the perception of social control (blue square 2) in order to directly reduce driving under the effect of substances (red square 3). However, this behavior is just an example of risky driving styles (red square 4) that could easily replace it in case of a successful suppression (light-green arrow 3.1) if a deep maladaptive motivation persists. The factors leading to the adoption of risky driving styles represent, in our opinion, the correct target for a comprehensive intervention (represented by dark-green arrows). This intervention should previously consider psychological, medical, and social variables predicting the adoption of risky driving styles (blue square 0). The final effect would concern the whole category of risky driving styles, including substance consumption (dark-green arrow 4.1).

## 5. Limitations

The present systematic review would have benefited from a meta-analysis, but sample heterogeneity and methodological variability in the assessment of substance consumption made this impossible. Further, the data aggregation adopted by some studies made some information useless for our purposes (e.g., not matching the incidence of road crashes with the positivity to substances). We contacted the corresponding authors of these studies asking for more detailed information, but they failed to provide it even after explicit request.

Finally, the strict inclusion criteria adopted during the selection of papers increased the qualitative comparability of results reported, albeit at the expense of a small pool of studies considered.

This all considered, the theoretical model proposed in the present paper—even if based on a systematic review of the scientific literature—is not meant to be conclusive and should be better confirmed in future research directly addressing the topic of interest (that is, the relation between substance consumption and road traffic accidents at work). The limitations highlighted in the present review should be considered in future studies centered on this research topic.

## 6. Conclusions

The present systematic review confirmed the significant impact of alcohol, drugs, and medicines on the probability of experiencing a work-related road traffic crash (RTC). While this relation could appear quite obvious, the included studies do not provide a deeper understanding of the reasons that lead so many people to consume substances in the workplace despite the growing awareness of the problem promoted by dedicated campaigns. Indeed, even the positive effects coming from effective control of this phenomenon could be vanished by new risky behaviors adopted to pursue maladaptive inner motivations. Preventive interventions could benefit from the contextualization of the deep reasons underlying substance consumption in the work environment.

From the descriptive statistics on positivity rates, it emerged that (1) the wide ranges found for each substance were reasonably attributable to methodological and sampling differences, and (2) medicines were the substances mostly associated with work-related RTC, which should increase the researchers’ interest in them (less than that in alcohol and drugs so far).

We strongly recommend that future studies will investigate the psychosocial variables possibly exposing workers to a higher risk of work-related road traffic crashes linked to the consumption of substances.

## Figures and Tables

**Figure 1 behavsci-12-00023-f001:**
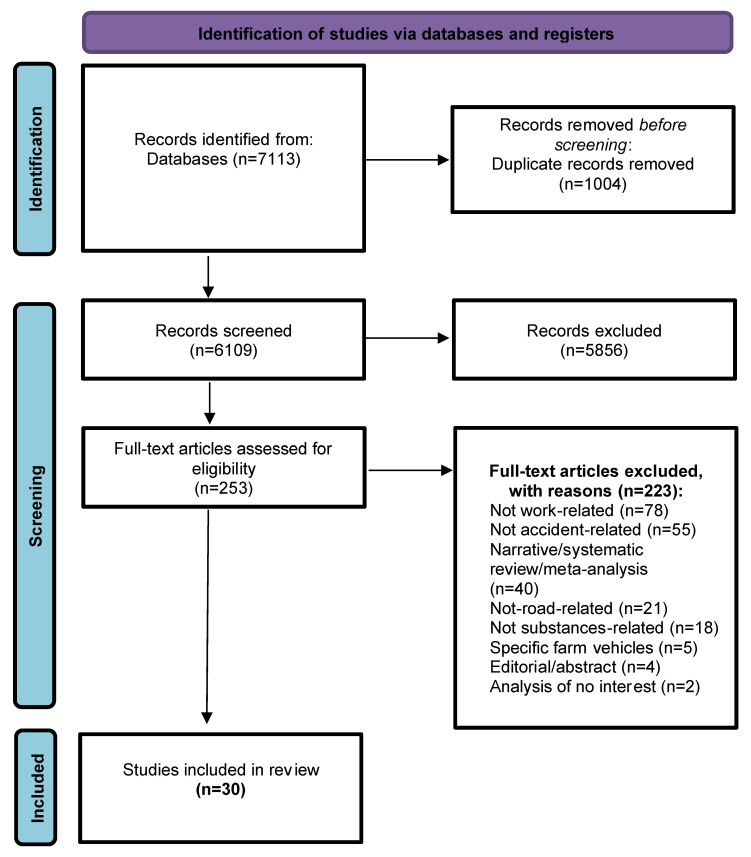
Flow Diagram.

**Figure 2 behavsci-12-00023-f002:**
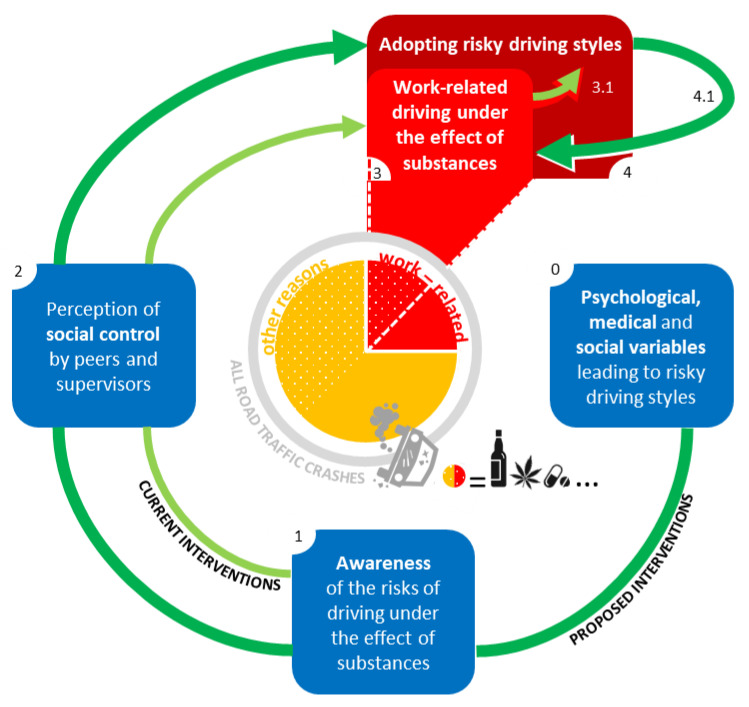
A model of intervention.

**Table 1 behavsci-12-00023-t001:** List of keywords for each area of interest.

Alcohol, Recreational Drugs, Medicines	Work	Road Traffic Crashes
alcohol	bus	accident
amphetamine	business driver	blameworthiness
analgesic	commercial driver	collision
antidepressant	commuting	crash
antihistamine	company car	crashes
anxiolytic	delivery worker	culpability
BAC	emergency vehicle	death
barbiturate	grey fleet	fatalities
benzodiazepine	heavy commercial vehicle	fatality
cannabinoid	itinere	incident
cannabis	job	injuries
cocaine	light commercial vehicle	injury
drink	lorry	near miss
drink-drive	occupational driver	road risk
drink-driving	private car	
drinking and driving	professional driver	
driving under the influence	public transport vehicle	
driving while intoxicated	taxi	
drug	trailer	
drunk	transport	
drunk driving	transportation	
drunk-driving	truck	
DUI	work	
DUID	workplace	
ethanol	work-place	
heroin	work-related	
hypnotic		
intoxicated		
medication		
narcotic		
opiate		
opioid		
polypharmacy		
psychoactive		
psychostimulant		
psychotropic		
sedative		
stimulant		
substance		
tranquilizer		

**Table 2 behavsci-12-00023-t002:** Participants, Interventions, Comparisons, Outcomes, and Study Design (PICOS).

Parameters	Inclusion Criteria	Exclusion Criteria
Participants	Healthy humans; Working population	Young people (<18 years) or older people (>65 years); Students, unemployed, or retired people; Subjects with chronic diseases.
Interventions	Use/abuse of alcohol during and/or near the use of vehicles on the road related to work; Use/abuse of psychotropic drugs during and/or near the use of vehicles on the road related to work; Use/abuse of drugs during and/or near the use of vehicles on the road related to work;	Use of very specific vehicles (e.g., trains, off-road vehicles, tractors, quads, etc.).
Comparisons	Any comparison; Stratification of the population according to age, gender, marital status, vehicle, years of experience, distance travelled, workload, night shifts, level of education, commuting.	Driving simulation studies.
Outcomes	Prevalence and characterization of alcohol/psychotropic drugs/drugs related traffic accidents during or close to working hours.	Injuries at work not related to alcohol/psychotropic drugs/drugs; Alcohol/psychotropic drugs/drugs related accidents that occurred outside or outside of working hours; Accidents related to alcohol/psychotropic drugs/drugs that did not occur on the road; Alcohol/psychotropic drugs/drugs related health problems; Use of alcohol/ psychotropic drugs/drugs AFTER an accident at work; Violence, homicide, suicide, or overdose in the workplace related to alcohol/psychotropic drugs/drugs use.
Study Design	Original studies: longitudinal, cross-sectional, randomized controlled, pre-post.	Case reports, narrative reviews, systematic reviews and meta-analyses; Lack of rigorous description of experimental methodology;

**Table 3 behavsci-12-00023-t003:** Retrieved studies and their main outcome.

Study	Country (Region)	Time Period	Data Source	Study Design	Substance(s) Investigated	Consumption Assessment	Sample(s) Features	Positivity Rates (N)	Synthesis of Main Findings
Asefa et al., 2015	Ethiopia (Mekelle)	2014	Recruitment from a representative population of taxi drivers	Cross-Sectional	Alcohol	Subjective (semi-structured questionnaire)	N = 712 taxi drivers	/	Self-reported history of alcohol use was an independent predictor of W-R RTC
Bacchieri et al., 2010	Brazil (Pelotas)	2006	Recruitment from a representative population of cycling commuters	Cross-Sectional	Alcohol	Subjective (semi-structured survey)	N = 1133 cycling commuters	/	Riding right after alcohol consumption was a risk factor only if considered together with other extremely imprudent behaviors
Bamberger and Cohen, 2015	Israel	2015	Recruitment of a random sample of employees from 8 transportation enterprises	Cross-Sectional	Alcohol	Subjective (AUDIT)	N = 227 commercial drivers (truck or bus)	/	Severity of alcohol misuse and number of accidents reported in the past year were significantly related
Boufous and Williamson, 2006	Australia (New South Wales)	1998–2002	Traffic Accident Database System (TADS), Workers’ Compensation Scheme Statistics (WCSS)	Cross-Sectional	Alcohol	Objective (illegal alcohol level)	N = 13,124 drivers injured/dead from W-R RTC	1.36 % (N = 179)	Over a 5-year period, an illegal alcohol level was found in a minority of workers driving on duty or during commuting
Chu, 2014	Taiwan	2005–2011	Taiwan’s National Police Accident Reports	Cross-Sectional	Alcohol	Objective (BrAC)	N = 1286 freeway high-deck buses involved in RTC	7.07 % (N = 91)	70% (N = 64) of 91 drivers reported as drunk were involved in fatal or injurious RTC
Karakus et al., 2015	Turkey (Izmir)	2010–2011	Izmir Forensic Medicine Group Presidency database	Retrospective Cross-Sectional	Alcohol	Objective (BAC)	N = 33 drivers involved in non-fatal W-R RTC	3.03 % (N = 1)	Comparing two different BAC limit, the highest significantly increased risk of non-fatal accident; the lowest did not
Sam et al., 2018	Ghana	2011–2015	National Accident Database	Cross-Sectional	Alcohol	Objective	N = 33,694 bus and minibus involved in RTC	1.87% (N = 630)	Drivers who tested positive for alcohol were more likely to have a more severe RTC
Smailović et al., 2019	Serbia	2016	National Traffic Accident Database	Cross-Sectional	Alcohol	Objective (BAC)	N = 3335 truck drivers involved in RTC N = 825 bus drivers involved in RTC	32.6% (N = 1087) 42.67% (N = 352)	Among commercial drivers involved in W-R RTC, positivity to alcohol was found in about 1/3
Thiese et al., 2015	USA	2015	Recruitment of a random sample of truck drivers	Cross-Sectional	Alcohol	Subjective (computerized questionnaire)	N = 797 truck drivers	/	Alcohol use significantly increased the risk of W-R RTC among truck drivers
Thygerson et al., 2011	USA (Utah)	1999–2005	Police crash reports and hospital inpatient and emergency department records	Cross-Sectional	Alcohol	Objective	N = 2330 workers who accessed the emergency department because of RTCN = 235 workers who were hospitalized because of RTC	1% (N = 31) 3% (N = 7)	W-R RTC were associated with a higher severity of prognosis and with a higher fatality rate
Chen et al., 2020	USA (Los Angeles)	2010–2018	US Statwide Integrated Traffic Records System (SWITRS)	Cumulative link mixed model	Alcohol	Objective	N = 21,258 truck drivers	7.26% (N = 1544)	Among various risky driving behaviors, alcohol consumption independently and significantly increases severity of W-R RTC
Konlan et al., 2020	Ghana (Adidome)	2018	Recruitment from a sample of commercial motorcyclists	Descriptive cross-sectional	Alcohol	Subjective (questionnaire)	N = 114 commercial motorcyclists	/	A history of alcohol use was associated with a higher prevalence of W-R RTC
French and Gumus, 2021	USA and Puerto Rico	2004–2012	Fatality Analysis Reporting System (FARS) database	Longitudinal	Alcohol	Objective (BAC)	N = 1800 traffic fatalities	2.1% (N = 38)	Alcohol was one of the factors increasing the number of fatalities due to W-R RTC during prosperous times
Mitchell et al., 2014	Australia (New South Wales)	2001–2011	Admitted Patient Data Collection (APDC)	Retrospective analysis	Alcohol	Objective (BAC)	N = 3888 car drivers and motorcyclists involved in RTC	1.77% (N = 69)	Risky behaviors (alcohol assumption) were more common in non-W-R journeys than W-R journeys
Poku et al., 2020	Ghana (Kintampo North Municipality)	2017	Recruitment from a sample of commercial vehicle drivers from Driver and Vehicle Licensing Authority (DVLA)	Cross-Sectional	Alcohol	Subjective (semi-structured questionnaire)	N = 126 commercial drivers involved in at least one RTC	/	Alcohol use significantly increased the risk of W-R RTC among commercial drivers
Brodie et al., 2009	Australia (Victoria)	1999–2007	Victorian State Coroner’s Office	Cross-Sectional	Alcohol, drugs (stimulants and cannabis)	Objective (BAC, toxicological screen)	N = 61 truck drivers killed in an RTC	Alcohol: 1.64 % (N = 1) Drugs: 16.3 % (N = 10)	Among heavy vehicle (≥ 4.5 tons) drivers, fatalities associated with consumption of drugs outnumbered those associated with consumption of alcohol
Holizki et al., 2015	Canada (British Columbia)	2003–2007	Workers’ Compensation Board of British Columbia	Cross-Sectional	Alcohol, drugs	Objective (toxicological screen)	N = 71 workers’ compensation claims for traumatic fatalities	Alcohol: 5.6 % (N = 4) Drugs: 18.3 % (N = 13) Alcohol and Drugs: 8.45% (N = 6)	Injurious and fatal crashes were more frequently associated with drugs than with alcohol or with a combination of the two Fatality rates were higher for small businesses (9.7) than for larger businesses (2.7)
Lambrechts et al., 2019	Belgium	2015–2016	Recruitment from a sample of the Belgian working population	Cross-Sectional	Alcohol, drugs	Subjective (AUDIT-C)	N = 4197 workers who used alcohol and 403 who used drugs, in the last year	/	Workers were found to consume alcohol more frequently than drugs; however, ratios were overturned when considering the prevalence of RTC
Rudisill et al., 2019	USA (West Virginia)	2000–2017	Fatality Analysis Reporting System (FARS) database	Cross-Sectional	Alcohol, drugs	Objective (BAC, toxicological screen)	N = 209 workers fatally injured in RTC	Alcohol: 3.82 % (N = 8) Drugs: 13.39% (N = 28)	The odds of being involved in a work-related fatal collision were predicted by alcohol Alcohol and drugs are less used during work-related journeys, but this reduction is halved for drugs
Yuan, 2021	USA and Puerto Rico	2012–2016	Fatality Analysis Reporting System (FARS) database	Partial Proportional Odds	Alcohol, drugs	Objective (BAC, toxicological screen)	N = 15,506 truck drivers	Alcohol: 2.4% (N = 372) Drugs: 1.2% (N = 186)	Among truck drivers with high risk of driving violations and high historical crash records, alcohol and drugs were significantly associated with the RTC severity
Li et al., 2020	USA (Texas)	2011–2015	Texas Crash Records Information System (CRIS)	Mixed Logit Model	Alcohol, drugs	Objective	N = 85,184 large truck involved in RTC	Alcohol: 1.29% (N = 1103) Drugs: 0.6% (N = 305)	Consumption of alcohol was more frequent than that of drugs, but ratio between serious and non-serious RTC was higher for drugs (1:3) than alcohol (1:4).
Liu and Fan, 2020	USA (North Carolina)	2005–2013	Highway Safety Information System (HSIS)	Mixed Logit Model	Alcohol, drugs	Objective	N = 7976 rear-end RTC involving large trucks	Alcohol and/or drugs: 0.45% (N = 36)	Driving under the influence of alcohol or drugs significantly increases the injury severity of RTC
Papalia et al., 2012	Italy	2011- 2012	Recruitment from a sample of employees of an urban and suburban transport company	Cross-Sectional	Alcohol, medicines (antihistamines and benzodiazepines)	Subjective (questionnaire)	N = 253 workers of an urban/extra-urban transport company	/	Use of antihistamines and benzodiazepines was found to significantly correlate with the risk of RTC
Bourdeau et al., 2021	France	2005–2015	Police reports (PRs)Bulletins d’Analyse des Accidents Corporels (BAAC)Système National d’Informations Inter Régimes de l’Assurance Maladie	Logistic Regression model	Alcohol, medicines (10 classes)	Objective (medicines prescriptions)	N = 21,490 workers involved in an injurious RTC during a W-R mission N = 43,012 commuters involved in an injurious RTC	15% (N = 3213) 17.9% (N = 7689)	Among the ten classes of medicines investigated, the higher risk of W-R RTC was associated with the assumption of: antiepileptics, psycholeptics or psychoanaleptics for the commuters; antiepileptics, other nervous system drugs or psychoanaleptics for the drivers on a work-related mission.
McNeilly et al., 2010	Australia (Victoria)	2001–2006	National Coroner’sInformation System database	Retrospective, observational, cross-sectional	Alcohol, drugs, and medicines	Objective (BAC, toxicological screen)	N = 64 worker and commuter deaths with positive toxicological screening	Alcohol: 14% (N = 9) Drugs: 21.33 % (N = 16) Medicines: 43.75% (N = 28)	Medicines are the substances more frequently associated with fatal W-R RTC
Qi et al., 2013	USA (New York State)	1994–2001	New York StateStatewide Work Zone Safety Inspection Program database	Cross-Sectional	Alcohol, drugs, and medicines	Objective	N = 2481 rear-end RTC	Alcohol, drugs, and medicines: 1.45% (N = 36)	Consumption of substances was associated with higher severity of W-R RTC
Gates et al., 2013	USA (Puerto Rico and D.C.)	1993–2008	Fatality Analysis Reporting System (FARS) database	Cross-sectional	Drugs (stimulant use)	Objective (toxicological screen)	N = 10,190 truck drivers involved in fatal RTC and tested for stimulant use	3.7 % (N = 372)	The use of stimulant was found to increase the risk of performing an unsafe driving action in a fatal crash (AOR = 1.78; 95%CI = 1.41–2.26)
Wadsworth et al., 2006	U.K. (Wales)	2001	Recruitment of a random sample from the electoral registers of Cardiff and Merthyr Tydfil	Cross-Sectional	Drugs (Cannabis)	Subjective (postal questionnaire survey)	N = 2801 W-R RTC	/	Cannabis use tripled the risk of injury
Khoshakhlagh et al., 2019	Iran (Teheran)	2011–2016	Recruitment from a random sample of Iranian truck and bus drivers (selected during annual healthy job visit)	Cross-Sectional	Medicines	Subjective (Specially designed questionnaire)	N = 323 truck and bus drivers	/	The consumption of two medicines significantly increased the incidence of RTC: Gemfibrozil (used to reduce cholesterol) and Glibenclamide (used to treat type 2 diabetes)
Reguly et al., 2014	USA (Puerto Rico and D.C.)	1993–2008	Fatality Analysis Reporting System (FARS) database	Case-control cross-sectional	Medicines (opioids analgesics)	Objective (toxicological screen)	N = 10,190 truck drivers tested for drugs	1.03 % (N= 105)	Male truck drivers using opioid analgesics had greater odds of committing unsafe driver actions

**Table 4 behavsci-12-00023-t004:** The main descriptive statistics concerning the positivity rates.

Substances	Weighted Average 95%CI	Range (Minimum-Maximum)
Alcohol	3.02% (2.21–4.04%)	1.29–42.67%
Drugs	0.84% (0.30–1.84%)	0.6–21.33%
Medicines	14.8% (10.74–19.8%)	1.03–43.75%

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
