# Peer review of "A Systematic Review on the Role of Substance Consumption in Work-Related Road Traffic Crashes Reveals the Importance of Biopsychosocial Factors in Prevention"

_behavsci, 2022, doi:10.3390/bs12020023_

Round 1

Reviewer 1 Report

The reviewed paper aims to systematically review the scientific literature concerning the association between the use or abuse of alcohol, drugs and medicines and work-related accidents. The authors tried to identify the root causes of this relationship. 

The manuscript is well structured and the presented material is clear and easy to understand. The investigated problem is significant and the paper managed to cover to a large extent the researches so far. The authors have accomplished the objective and even it is well known that substance use significantly increases the risk of road traffic crashes this systematic review is valuable enough. Most important it becomes clear that despite the there is still a gap in the understanding of the reasons that lead to substances consumption at work. The authors also propose an intervention model and note possible guidelines for future work.

Figures and tables represent the information appropriately and are easy to understand. Only two comments here: In figure 1 the paragraph markers and other characters like spaces are not hidden which visually complicates the perception. In Table 3 the first column is named “Partecipants” which I believe is "Participants". 

I have two suggestions for the authors:

First, I think it is important to make the difference between the use and the abuse of alcohol, drugs and medicines in relation to RTC. It is not clear how many of the studies examine the use and how many of them examine the abuse. If such information is available probably it should be pointed out in the discussion section. 

And second, I also believe that there is a difference between individuals using the investigated substances due to their specifics. While it is a personal choice to take or not drugs or to drink and then drive, it is not a choice to take or not some medicines like insulin for example. In the first case, the decision doesn’t concern the life-supporting need for the individual which is the situation with some diseases (sure not chosen from the individual). My suggestion is to highlight this in the discussion section when summarizing the results about the medicines.  

Congratulations to the authors for the work done! 

Reviewer 2 Report

Thank you for the opportunity to review this interesting paper. I only have minor comments.

1) I really enjoyed the concise writing up until line 467. In my perception, it fizzles out here.  While I do see the importance of the bigger picture, I am not convinced that the wording is correct here. You investigated all types of RTC not only at-fault ones, so I do not see why you mention "misbehaviors" in line 471. Also, I think the classic "blaming personality" is not appropriate at all. Whenever we psychologists investigate "personality" or "attitudes", they are far less predictive than actual observable behavior. Please be a little less populistic in your last paragraph.

2) You excluded 40 papers that are already reviews. Could you integrate their results somehow in your paper and use your data as an integrating update on the existing literature? That would enhance the worth of your work very much.

3) in line 392 there is an "." before the reference, please delete.

Reviewer 3 Report

This study tries to provide a systematic review on the relationship between substance consumption (i.e., alcohol, drugs, and medicines) and work-related traffic accidents. This topic is very interesting and has its contributions, and the authors have paid a lot of effort in relevant article match and selection. However, due to some reasons, this study only focuses on comparing the results that are directly extracted from previous studies, without using statistical methods for further analyses. This leads to that the current findings in this study cannot draw a compressive and in-depth conclusion, such as whether these impacting factors are significant, what the detailed quantified associations are, etc., after combining the data from all these selected previous studies. Many important pieces of information need to be supplemented, which could help make this study more meaningful.

Reviewer 4 Report

This manuscript has serious defects.

  1. The background, purpose, methodology and main findings and significance of the study should be clearly stated.
  2. It is impossible to understand from the Introduction why the authors had to carry out this research. Why do you choose alcohol/drugs/medicines? What’s the limitation of current studies? Since there is chapter 1.1 (4.1), then why is there no chapter 1.2 (4.2)?
  3. Table 2 should be revised, and many words are repetitive.
  4. Chapters 3.2 and 3.3, the authors just present the main results of the retrieved articles; this is too sample and not enough. The authors should summarize the characteristics and results of these papers, not just state the results of others studies.
  5. For the discussion, the authors should state the relationship between your research findings and other scholars, as well as explain your contribution to the current studies.
  6. There are some grammatical errors in the article. Please check the English language throughout the paper.

Reviewer 5 Report

The main purpose of the review paper was to systematize the scientific literature linking the assumption of alcohol, drugs, and medicines with work-related road traffic crashes (RTC), which might be benefit in preventing road traffic crashes relative to substance consumption. This is interesting and valuable review.

   The Preferred Reporting Items for Systematic Reviews and Meta-Analyses (PRISMA) guidelines are employed to carry out the systematic review, and the search engines such as PubMed, Google, and Google Scholar are used. 30 articles have finally been picked out from the pool of 6,109 abstracts after 3 stages of filtrate. Some bright spots have been found:

  1. A history of substance abuse might be an independent predictor of work-related RTC.
  2. Preventive interventions could benefit from the contextualization of the deep reasons underlying substance consumption in the working environment.

   This is a perfect review paper and is worth to be published. However, there are some type mistakes in the paper which should be removed, such as page 7, Figure 1(Flow Diagram) have many strange symbols.

Round 2

Reviewer 3 Report

The authors have made significant changes to improve the quality of their study. I'd like to recommend it to be published in this journal.

Reviewer 4 Report

My comments have been addressed.